# Multiple Nucleocapsid Structural Forms of Shrimp White Spot Syndrome Virus Suggests a Novel Viral Morphogenetic Pathway

**DOI:** 10.3390/ijms24087525

**Published:** 2023-04-19

**Authors:** Hui-Ju Huang, Sen-Lin Tang, Yuan-Chih Chang, Hao-Ching Wang, Tze Hann Ng, Rees F. Garmann, Yu-Wen Chen, Jiun-Yan Huang, Ramya Kumar, Sheng-Hsiung Chang, Shang-Rung Wu, Chih-Yu Chao, Kyoko Matoba, Iwasaki Kenji, William M. Gelbart, Tzu-Ping Ko, Hwei-Jiung (Andrew) Wang, Chu-Fang Lo, Li-Li Chen, Han-Ching Wang

**Affiliations:** 1Institute of Marine Biology, National Taiwan Ocean University, Keelung 20224, Taiwan; 2Institute of Biological Chemistry, Academia Sinica, Nankang, Taipei 115, Taiwan; 3Biodiversity Research Center, Academia Sinica, Taipei 11529, Taiwan; 4International Center for the Scientific Development of Shrimp Aquaculture, National Cheng Kung University, Tainan 701, Taiwan; 5The Ph.D. Program for Translational Medicine, College of Medical Science and Technology, Taipei Medical University and Academia Sinica, Taipei 110, Taiwan; 6Graduate Institute of Translational Medicine, College of Medical Science and Technology, Taipei Medical University, Taipei 110, Taiwan; 7Department of Biotechnology and Bioindustry Sciences, National Cheng Kung University, Tainan 701, Taiwan; 8Temasek Life Sciences Laboratory, National University of Singapore, Singapore 117604, Singapore; 9Department of Chemistry and Biochemistry, San Diego State University, San Diego, CA 92182-1030, USA; 10Institute of Oral Medicine, National Cheng Kung University, Tainan 701, Taiwan; 11Department of Physics and Graduate Institute of Applied Physics, National Taiwan University, Taipei 10617, Taiwan; 12Molecular Imaging Center, National Taiwan University, Taipei 10617, Taiwan; 13Protein Synthesis and Expression, Institute for Protein Research, Osaka University, Osaka 565-0871, Japan; 14Life Science Center for Survival Dynamics, Tsukuba Advanced Research Alliance (TARA), University of Tsukuba, Tsukuba 305-8577, Japan; 15Department of Chemistry and Biochemistry, University of California, Los Angeles, CA 90095-1569, USA; 16Center of Excellence for the Oceans, National Taiwan Ocean University, Keelung 20224, Taiwan

**Keywords:** white spot syndrome virus, large dsDNA virus, cryogenic electron microscopy, nucleocapsid

## Abstract

White spot syndrome virus (WSSV) is a very large dsDNA virus. The accepted shape of the WSSV virion has been as ellipsoidal, with a tail-like extension. However, due to the scarcity of reliable references, the pathogenesis and morphogenesis of WSSV are not well understood. Here, we used transmission electron microscopy (TEM) and cryogenic electron microscopy (Cryo-EM) to address some knowledge gaps. We concluded that mature WSSV virions with a stout oval-like shape do not have tail-like extensions. Furthermore, there were two distinct ends in WSSV nucleocapsids: a portal cap and a closed base. A C14 symmetric structure of the WSSV nucleocapsid was also proposed, according to our Cryo-EM map. Immunoelectron microscopy (IEM) revealed that VP664 proteins, the main components of the 14 assembly units, form a ring-like architecture. Moreover, WSSV nucleocapsids were also observed to undergo unique helical dissociation. Based on these new results, we propose a novel morphogenetic pathway of WSSV.

## 1. Introduction

The white spot disease (WSD) of shrimp has been on the list of diseases notifiable to the World Organization for Animal Health (WOAH, formerly known as OIE) since 1997. Currently, WSD is still a major concern for the global shrimp industry [1]. The cause is white spot syndrome virus (WSSV), with a double-stranded DNA (dsDNA) genome of 305~307 kbps [2,3,4]. WSSV can be classified as a giant virus, as it satisfies the criteria of enclosing large dsDNA (≥300 kbps, and at least one of its capsid dimensions is ≥ 200 nm). The genome of the WSSV Taiwan isolate 1 (NCBI Acc. No.: AF440570) is ~307 kbp and contains 532 putative open reading frames (ORFs) [5]. Some viral proteins encoded by these ORFs are implicated as having unique functions, including anti-apoptosis, histone binding, and viral replication [5]. It is well-accepted that the WSSV virion is the largest, ovoid-to-bacilliform-enveloped virus with a tail-like extension [2,6,7]. Based on its uniqueness and the tail-like extension of the WSSV virion particle, the taxonomic position of WSSV has been assigned to a new family of *Nimaviridae* (Nima in Greek means thread) as the only member of the genus Whispovirus [5].

The morphogenesis of the WSSV virion occurs completely in the nucleus, generating a viral particle with three layers (a lipid-containing envelope [LEN], tegument [TG], and nucleocapsid [NC]) that enclose the WSSV genome[s] [8]. The outer lipid-containing envelope of the WSSV virion contains >34 structural proteins [8,9,10], notably the major component WSSV VP28, with critical roles in recognition, entry, and assembly [11]. The tegument layer of WSSV is composed of at least six viral proteins that are loosely associated with the envelope and the nucleocapsid [8]. WSSV VP26 and VP24 are two major tegument proteins that interact with the envelope protein, VP28, and function as anchors, linking the outer lipid-containing envelope and inner nucleocapsid [12]. 

Moreover, a stacked ring-structured nucleocapsid encloses the WSSV genomic DNA [6,13,14,15]. A shorter, fatter, rounder form of the nucleocapsid is present in the tightly packed intact virion, whereas a thin, rod-shaped, relaxed form of the nucleocapsid, with stacked rings, is observed when the nucleocapsids are unwrapped from the envelope [6,13,14,15]. To our knowledge, at least nine viral proteins contribute to the assembly of WSSV nucleocapsids [8,9,10]. Based on immunoelectron microscopy (IEM), viral protein WSSV VP664 (664 kDa), the largest known viral structural protein, has been proposed to form the unique stacked ring-like structure of the nucleocapsid [15]. Based on conventional transmission electron microscope (TEM) imaging, the intact enveloped virions ranged from 210 to 380 nm long and 70 to 167 nm wide [5,13,14].

Although IEM and TEM provided preliminary images of the morphology of the WSSV virions, most do not have the widely accepted shape, i.e., an ovoid-to-bacilliform-enveloped virus with a tail-like extension. Furthermore, the common depiction of WSSV nucleocapsids, a thin rod shape with stacked rings or conspicuous vertical helices, has not been clearly defined [14,15,16]. Therefore, WSSV morphology requires further clarification. In this study, we used TEM and cryogenic electron microscopy (Cryo-EM) to document the morphology of the WSSV virion and nucleocapsid in detail and to suggest a potential pathway for DNA packaging into nucleocapsids. We also used cryo-electron tomography (Cryo-ET) and single particle Cryo-EM to resolve structural issues of the WSSV nucleocapsid. Lastly, using a Cryo-EM map, we proposed a new model for the assembly units and morphogenesis of WSSV. Based on these findings, the morphology of WSSV virions and nucleocapsids differed substantially from the current understanding. Therefore, we revised the general morphology of an intact mature WSSV virion and nucleocapsid and proposed a novel WSSV morphogenesis.

## 2. Results and Discussion

### 2.1. TEM and Cryo-EM Analysis Showed Differential Shapes of Enveloped WSSV Virions and Nucleocapsids

To investigate the structures of the WSSV virion and nucleocapsid using TEM and Cryo-EM, crayfish were challenged with WSSV inoculum, and the moribund crayfish were collected to purify the intact virion and nucleocapsid by differential solutions and ultracentrifugation. The intact virion and nucleocapsid were prepared for TEM by negative staining, and for Cryo-EM by plunge-freezing into liquid nitrogen. Representative micrographs captured by the TEM and Cryo-EM of enveloped WSSV virions and nucleocapsids are shown (Figure 1). When observed with TEM, there appeared to be two forms of fully mature virions: (1) an intact enveloped, short sausage-shaped virion (so-called body) without a tail-like appendage; and (2) an intact enveloped virion with a short sausage-shaped body and a thread-shaped, tail-like extension (Ext.; Figure 1A). Some of the tail-like extensions were flexible (Figure 1Ai), whereas others had a bubble-like protrusion (Figure 1Aii). Dropped lipid-containing envelopes (Dro-LEN) and unwrapped nucleocapsids (Uw-NC) may be derived from apparently mature virions. The length of the Uw-NC was much longer than that of the intact enveloped virions (Uw-NC: 370 ± 20 nm; intact enveloped virions: 305 ± 25 nm; Appendix A) and much thinner (Uw-NC: 80 ± 10 nm; intact enveloped virions: 110 ± 10 nm; Appendix A). After the nucleocapsids were purified from the virion samples treated with 0.1% Triton X-100 and 0.5 M of NaCl, some naked nucleocapsids without genomic DNA had thin, striped tube-like stick structures with two distinct ends: an open end and a closed basal end (Figure 1B). 

Interestingly, based on the Cryo-EM observations (Figure 1C), there were two forms of mature virions: (1) an intact enveloped virion with a shorter, oval shape (Figure 1Ci); and (2) an intact enveloped virion wrapped with a loose LEN (Figure 1Cii,1Ciii). The nucleocapsid in the intact enveloped virion with the oval shape was much shorter, thicker, and compressed (Figure 1Ci) compared to those in the intact enveloped virions wrapped with a loose LEN (Figure 1Cii,Ciii). In addition, compared to the TEM, the size of the intact virions without the tail-like extension seen in the Cryo-EM was much longer and thicker (Cryo-EM: 353 ± 55 nm and 179 ± 22 nm; TEM: 305 ± 25 nm and 110 ± 10 nm; Appendix A). Some unknown aggregations were present in the lumen between the nucleocapsid and the loose envelope (Figure 1Cii). Based on our previous report [8], perhaps this fibril-like unknown aggregation is a complex of WSSV genomic DNA and viral DNA-binding proteins that might be folded and packed into the nucleocapsid. Moreover, the cap end of the nucleocapsid often seemed to direct toward the lumen of the loose envelope, where the WSSV genomic DNA-like aggregation was present (Figure 1Cii). This type of WSSV virion was also observed in the nucleus of WSSV-infected shrimp cells during WSSV morphogenesis (Figure 1D).

Furthermore, the naked nucleocapsids observed with Cryo-EM (Figure 1E) were much longer and thicker than those observed with TEM (Cryo-EM: 450 ± 17 nm and 77 ± 5 nm; TEM: 370 ± 20 nm and 80 ± 10 nm; Appendix A). In addition, the purified nucleocapsids had varying segment numbers. More than 75% of the nucleocapsids were assembled with 16–18 segments, and the most common (>35%) segment number of the WSSV nucleocapsids was 17 (Appendix A). Unlike the naked nucleocapsids shown in Figure 1B, the intact naked nucleocapsids with WSSV genomic DNA were asymmetrical at both ends: the cap end had a sharp front with a narrow platform, whereas the basal end had a slight curve with a wider platform (Figure 1E). Based on our previous study [8] and other dsDNA virus studies [17,18,19,20], we inferred that the cap end functions as a portal cap (PC), a complex molecular motor through which the viral genome is translocated into the capsid during virion morphogenesis [20,21]. A very recent study by Sun et al. (2023) further suggested that the head (or the cap) of the WSSV nucleocapsid consists of a rotor-like structure, whereas the base has a disc-shaped structure [22].

Most importantly, under Cryo-EM observation and tissue sections by TEM, WSSV virions with and without thread-like, tail-like extensions were observed. The tail-like extensions in the TEM images corresponded to the loose LEN wrapped around the nucleocapsid (Figure 1Cii). During maturation, the nucleocapsid conformation changed from a rod shape to an oval shape, where the LEN wrapped tightly around the nucleocapsid. Based on the Cryo-EM images, most of the mature virions were shorter, oval-shaped structures (Figure 1Ci,F). Three distinct layers of the WSSV virion, a lipid-containing envelope (LEN), tegument (TG), and nucleocapsid (NC), were clearly observed (Figure 1F). The outer side of the LEN had a filamentous structure, proposed for attaching to host cells [14,23]. The nucleocapsid had two distinct ends, i.e., the PC and the base, (Figure 1F). Taken together, we concluded that the morphology of the intact WSSV virion should be redefined as a shorter oval shape without a tail-like extension. The WSSV virion with “an envelope surplus” (previously described as a “tail-like extension,” based on TEM images of WSSV) should be regarded as an immature intact type of WSSV as the term “tail-like extension” is inappropriate.

### 2.2. Three-Dimensional Volume Reconstruction of WSSV Nucleocapsid by Cryo-Electron Tomography (Cryo-ET)

To further investigate the two distinctive ends of the WSSV nucleocapsid (Figure 1E), cryo-electron tomography (Cryo-ET), an emerging imaging modality, was used to obtain three-dimensional (3D) constructions of intact naked WSSV nucleocapsids (Figure 2A). Similar to a baculovirus nucleocapsid, with its nipple-like portal cap and claw-like basal end [24], the resulting 3D construction of the WSSV nucleocapsid also had two distinctive ends: a two-layer portal cap (Figure 2B,C) and a concave disk-shaped basal end (Figure 2D,E). The width of the second layer of the WSSV NC portal cap was ~55% of that of the first layer, whereas the width of a concave disk-shaped basal end was 46 ± 2 nm. Both ends were narrower than the middle part of the naked nucleocapsid. 

### 2.3. Proposed WSSV Genomic DNA Translocation through the Cap End of WSSV Nucleocapsids

For the non-enveloped single-stranded (ss) RNA virus, Tobacco Mosaic Virus (TMV), a two-layer coat protein (CP) disk is formed to initiate genome packaging, which starts at a special sequence of the genome, and then these disks are dislocated to form a virus with lock washer-shaped rings [25]. This spontaneous packaging was in contrast to the dsDNA viruses, such as HSV type 1 and baculovirus, where empty capsids are formed first, with DNA encapsulation subsequently accomplished by translocation through “motor” viral proteins [19,20,21]. From our TEM observations (Figure 3A,B), we propose that WSSV uses a similar active mechanism of translocation to achieve genome packaging. In general, an open portal cap end and a closed basal end were observed within the naked nucleocapsid without WSSV genomic DNA. The naked WSSV nucleocapsid sheath was a “long and thin” striped tube with an open portal cap end and a relatively restricted, closed basal end (see Figure 3Aa). The static images (Figure 3Aa–d) were arranged based on (i) the stages of cap formation; (ii) the change in the shape of the nucleocapsid from thin-rod shaped to shorter fat-rod shaped; and (iii) the amount of DNA packaged in the nucleocapsid (successive increase in DNA amount). Accompanied by DNA packaging into the nucleocapsid through the opened portal cap end, the portal cap-like structure was formed gradually in the apical direction (see Figure 3Ab,c). In the end, the portal cap of the nucleocapsid was closed, with the assembled intact nucleocapsid being swollen compared to the start (see Figure 3Ad).

For a close look at the cap formation of the WSSV nucleocapsids (Figure 3Aa–d), enlarged images of the cap ends were examined (Figure 3B). The possible scenario of the cap formation is proposed below: at the beginning, near the segment equivalent to the first of the mature nucleocapsid, some fiber-like structures seemed to surround the open portal cap region of the empty nucleocapsid (Figure 3Ba). The part of the nucleocapsid above this segment, but below the fiber-like structures, started to shrink, and the fiber-like materials seemed to form several unknown globin-like (GLO) structures (Figure 3Bb). After WSSV genomic DNA was packed into the nucleocapsid, the two layers of the portal cap were formed, whereas the assembled globin-like structure (ASS. GLO) was still observed (Figure 3Bc). Eventually, WSSV genomic DNAs seemed to fill in the nucleocapsid to constitute a mature nucleocapsid, and the portal cap was also formed (Figure 3Bd). The above description regarding the DNA translocation event of WSSV nucleocapsids was also clearly demonstrated in Figure 3C; the DNA-like filament was translocated through the portal cap end of the nucleocapsid. Based on other DNA viruses, e.g., phage, baculovirus, and HSV, the portal structure through which the viral genome was packaged acted as an exit during genome ejection, and the last-packaged end of the genome ejected first [17,18,19,20,21]. 

However, we note that these nucleocapsids from crayfish were purified and handled extensively prior to staining and imaging. Consequently, the proposed process of WSSV genomic DNA packaging (i.e., Figure 3) was based on collective observations of nucleocapsids with different shapes, rather than true examples of intermediate states of the viral lifecycle.

### 2.4. Cryo-EM Map of the Middle Part of the WSSV Nucleocapsid (M-NC) Revealed a Unique Architecture with C14 Symmetry

As described above and shown in Appendix A, the intact WSSV virions, even the nucleocapsids without genomic DNAs, exhibited pleomorphism when electron microscopy was used to study the morphology of the whole virus and nucleocapsid [8,15]. Consequently, it was not possible to average the images of either (the whole virion or whole nucleocapsid) to gain more detailed information. Nevertheless, Cryo-EM examinations provided a good opportunity to assess the WSSV nucleocapsids in terms of potential assembly units and processes.

Instead of the intact WSSV virion, the middle part of the WSSV naked nucleocapsids (M-NC) was selected for the Cryo-EM study, due to its consistency (Figure 4A). Cryo-EM single-particle reconstruction was used to determine the overall structure of the middle part of the WSSV nucleocapsids. A Cryo-EM map of WSSV M-NC was eventually reconstructed, using ~30 K collected images. The resolution of the resulting map was 5.5 Å, insufficient to build a reliable, atomic model for WSSV M-NC. However, the Cryo-EM map was adequate to describe potential assembly units and the process of WSSV M-NC. There were two types of WSSV M-NC, narrow and wide (Figure 4B), of which ~58% were narrow. The inner and outer widths of the narrow type of WSSV M-NC were 475.4 and 681.8 Å, respectively, whereas they were 504.4 and 736.1 Å for the wide type (Figure 4B). 

Based on TEM, one WSSV nucleocapsid ring was described as comprising 36 triangular bodies and 18 rhomboid bodies [14]. By analyzing the asymmetry of the Cryo-EM map, we substantially revised the WSSV nucleocapsid ring structure: both WSSV M-NC types were assembled by the same 14 nucleocapsid units (NUs) (Figure 4B). Based on the characteristics of our Cryo-EM map, we separated each NU into flower (F) and stem (S) domains (Figure 5, left panel). Based on pairwise comparisons, the two domains of narrow-type and wide-type NUs (n-NU and w-NU) appeared to have distinct conformations. The most notable difference was in the F domain. Specifically, the horizontal axis of this domain in the w-NUs was largely extended when compared to that of the n-NUs. The F domains of the n-NUs were 118 Å, shorter than the 142 Å width of the w-NUs (Figure 5). Extension of the horizontal axis of w-NUs is possible to result in a looser package for creating a larger capacity of WSSV nucleocapsid (Figure 4 and Figure 5). This structural difference of NUs may enable the thin, striped, tube-like, stick-structured naked nucleocapsids to undergo conformational change into oval-shaped WSSV nucleocapsids with DNA genome in the intact WSSV virion particle (Figure 1F). Furthermore, these 14 NUs seemed to form a ring-like architecture (Figure 5), implying these NUs may be assembled into filament-like structures and subsequently spatially arranged into a helix. Therefore, WSSV nucleocapsids would have helical symmetry. A proposed dissociation mechanism of WSSV nucleocapsids with helical symmetry is shown in Figure 6A,B. First, the dissociation of the WSSV nucleocapsid seemed to begin from the portal cap end, not from the concave disk-shaped basal end (Figure 6A). The WSSV nucleocapsids did not dissociate into individual rings but rather dissociated as 14 filament-like structures (Figure 6B), implying that the vertical packing of the NUs was stronger than the horizontal packing. As mentioned above, the F domains of the WSSV NUs were flexible (Figure 5). The F domain may induce a conformational change between n-NUs and w-NUs by a horizontal extension (Figure 5). We further hypothesize that an extra extension of F domains reduced the horizontal binding between WSSV NUs, causing a unique, 14-helical strand-like dissociation of the WSSV nucleocapsids.

Various conformations of WSSV M-NC may be related to stages of packaging WSSV nucleocapsids (Figure 3, Figure 4, Figure 5 and Figure 6). Our hypothesis is that, after synthesis, NUs form filament-like structures and are arranged into helically symmetrical stick-like nucleocapsids which initially organize into the narrow type, and subsequently, in response to the increasing volume of incoming genomic DNA, a conformational change in the F domain leads to the transformation of the narrow type into the wide type of WSSV nucleocapsids. A similar situation arises in the case of well-studied bacteriophage HK97, where the pre-formed “procapsid” into which the genome is being packaged “matures” into a larger, faceted, capsid after a certain fraction of the dsDNA has been loaded [26].

### 2.5. WSSV VP664 Is the Main Component of the 14 NUs of WSSV Nucleocapsid

WSSV nucleocapsids are formed by at least nine distinct WSSV structural proteins, ranging from 15~664 kDa [8,15]. VP664 is the major WSSV nucleocapsid protein and the largest viral structural protein known. Leu et al. (2005) [15] suggested that VP664 should be a major component in the formation of the unique stacked rings in the nucleocapsid. 

We propose a C14 symmetric structure of the WSSV nucleocapsid (Figure 4 and Figure 5), based on the 5.5 Å density map. To confirm the C14 structural model of WSSV nucleocapsids, although we did not dissociate the WSSV nucleocapsid with protease, sonication, or freeze-thaw methods, we noticed some ring-like structures associated with the portal cap end of the nucleocapsid (Figure 7A,B) as well as individual ring-like structures (Figure 7C). In the IEM images after applying an anti-N-terminal VP664 serum and a gold-labeled secondary antibody, gold particles were distributed around the external surface of the nucleocapsid with a periodicity that approximately matched the number of 14 NUs (Figure 7Ca–d). We inferred that VP664, at least its N terminal region, should be the major part composing the external surface of 14 NUs.

### 2.6. Proposed Model for WSSV Morphogenesis 

Viral morphogenesis can be broadly divided into the following phases: attaching to the receptor/co-receptor on the host cell’s surface, entry into the host cell, uncoating of the genome, viral gene expression, viral genome replication, virion assembly, and release. Although WSSV morphogenesis has been described and discussed [14,27,28,29], the TEM and Cryo-EM yielded novel insights into WSSV morphogenesis. Our new information included: (1) the WSSV nucleocapsid had C14 symmetry (Figure 8A); (2) the WSSV nucleocapsid with or without DNA had various conformations (Figure 8B, left panel); (3) the intact mature WSSV virion was devoid of a tail extension (Figure 8B, right panel); (4) the narrow-type WSSV nucleocapsid with the volume of 798.3 K nm^3^ (NC volume: 3.14 × [23.77 nm] 2 × 450 nm ≈ 798.3 K nm^3^) accommodated at least one copy of WSSV genomic DNA (along with its associated proteins such as VP15); and (5) the WSSV nucleocapsid structure was in a helical manner rather than individual rings (Figure 8C). These findings contribute to an improved understanding of WSSV morphogenesis. 

We incorporated our observations into the current knowledge by proposing a more detailed model for WSSV morphogenesis (Figure 9). The steps are as follows: (1) An infectious WSSV virion attaches to the receptor/co-receptor on the susceptible host cell with viral envelope proteins and enters the cell via endocytosis [30,31,32]. (2) The viral envelope fuses with the endosome, and the naked nucleocapsid is transported to the outer layer of the nucleus [33,34]. (3) The WSSV genome is released into the nucleus. (4) Once in the nucleus, the WSSV genes are transcribed and expressed [35]. (5) The WSSV genomes are replicated 6~12 h post-infection [33,36]. (6) Nucleocapsid proteins such as VP664 form the nucleocapsid [15]. The nucleocapsid displays C14 symmetry, and the nucleocapsid is in a helical manner. The virogenic stroma started forming vesicles that constitute the viral envelope [37]. The immature capsid can also be observed as an elongated rod-like structure in the virogenic stroma [37]. (7) The viral genome, chromatinized by viral DNA-binding proteins (e.g., VP15), was transported into a mature capsid through the portal cap of the nucleocapsid [38]. The nucleocapsids had two distinct conformations based on the presence or absence of DNA. The narrow-type nucleocapsid fits in at least one copy of the WSSV genome plus the DNA-binding proteins. (8) The envelope precursors are expected to begin to shrink and remove excess parts to fit the nucleocapsid. (9) New mature WSSV virions lack a tail-like extension and are completely assembled in the nucleus. (10) Mature virions are released from the disrupted cell to initiate another morphogenesis cycle in susceptible cells [2].

The three-dimensional information for WSSV naked nucleocapsids was obtained with Cryo-EM (Figure 4 and Figure 5) with major conformational changes in the WSSV nucleocapsids. However, relatively low resolution (~5.5 Å) and a lack of VP664 crystal structure limited assessing detailed molecular architecture. Regardless, taken together, our results improved and revised the current understanding of WSSV morphogenesis, opening new directions for research on this large dsDNA virus.

## 3. Materials and Methods

### 3.1. Preparation of WSSV Inoculum

The WSSV Taiwan isolate strain (NCBI accession No: AF440570), originally collected from WSSV-infected *Penaeus monodon* in Taiwan in 1994, was used in this study. The virus for the experimental inoculum was prepared from the hemolymph of moribund-specific pathogen-free white shrimp (*Litopenaeus vannamei*) infected with the WSSV Taiwan isolate. After dilution with phosphate buffer saline (PBS; 137 mM of NaCl, 2.7 mM of KCl, 10 mM of Na_2_HPO_4_, 2 mM of KH_2_PO_4_) and centrifugation (13,000× *g*, 5 min, 4 °C) to remove hemocytes, the WSSV inoculum stock was stored at −80 °C. 

### 3.2. Purification of WSSV Virions and Nucleocapsids

To obtain purified samples of WSSV virions and nucleocapsids, WSSV-infected tissue was collected and processed as described [8,10,15], with some modifications. Initially, healthy crayfish (*Procambarus clarkii*; 10~20 g body weight) were injected intramuscularly with 0.1 mL of diluted (1: 500 in PBS) virus stock. From 3~9 d later, the moribund crayfish were collected, their eyes and hepatopancreas removed, and the remainder stored at −80 °C until used.

For the intact WSSV virion purification, the frozen tissues were cut into small pieces, homogenized in freshly prepared TESP buffer (50 mM of Tris, 5 mM of EDTA, 1 mM of PMSF, 500 mM of NaCl, pH 8.5; 10 mL/g tissue) and centrifuged for 5 min at 3500× *g* at 4 °C. The supernatant was then collected for high-speed centrifugation at 30,000× *g* (Type 45 Ti rotor, Beckman, CA, USA) for 30 min at 4 °C. After centrifugation, the supernatant and pink layer above the white pellet were gently removed with a Pasteur pipette. The resulting white pellet was resuspended in TM buffer (50 mM of Tris, 5 mM of MgCl_2_, pH 7.5) and centrifuged at 3500× *g* for 5 min at 4 °C. These steps were repeated twice, and the final white pellet was suspended in TM buffer containing protease inhibitor (EDTA-free, Roche, Taipei, Taiwan) and stored at −80 °C until used. 

For the nucleocapsid purification, purified virion fractions, prepared as described above, were treated with 1% Triton X-100 and 0.5 M of NaCl and then incubated at room temperature for 30 min, with gentle shaking to remove the envelope and tegument layers. After centrifugation at 28,000× *g* (SW 40 Ti rotor, Beckman) for 20 min at 4 °C, the supernatant was removed and the pellet was rinsed gently with TM buffer. After washing, the pellet was suspended in TM buffer and treated with DNase at 37 °C for 20 min. Finally, the nucleocapsids were collected by removing the supernatant after another centrifugation at 28,000× *g* (SW 40 Ti rotor, Beckman, California, USA) for 20 min. The nucleocapsid pellet was suspended in TM buffer containing protease inhibitor and stored at −80 °C until used. 

The purity, quality, and quantity of the intact WSSV virions and nucleocapsids were verified by SDS-PAGE and Western blotting. The virus samples were also assessed by negative staining with phosphotungstic acid (PTA), uranyl acetate (UA), or uranyl formate (UF) and subsequently examined under a transmission electron microscope (H7500 [Hitachi, Tokyo, Japan] or JEOL JEM-1400 [JEOL Co., Ltd], Tokyo, Japan), or Tecnai F20 [FEI]).

### 3.3. Cryo-EM Sample Vitrification 

For the Cryo-EM, 4 µL of purified WSSV virions or nucleocapsids was applied onto glow-charged 200-mesh Quantifoil R2/1 holey carbon grids. These grids were blotted in 100% humidity at 4 °C for 3.5 s and then plunge-frozen into liquid ethane cooled by liquid nitrogen, using a FEI Vitrobot Mark IV (ThermoFisher Scientific, Taipei, Taiwan). The grids were stored in liquid nitrogen until used. 

### 3.4. Data Acquisition

The frozen, hydrated grids were maintained at near-liquid nitrogen temperatures (−176 to −180 °C) during the imaging. The images were collected on a Talos Arctica cryo-electron microscope (ThermoFisher Scientific, Taipei, Taiwan) operated at 200 keV. For the single-particle analysis, the low-dose condition was~48 e^−^/Å^2^. The images were captured at a magnification of 73,000× with a defocus range of −0.5 to −2.5 μm and recorded with a Falcon III detector (ThermoFisher Scientific, Taipei, Taiwan). For the cryo-electron tomography, a series of images were collected automatically from −60° to +60° at 2° intervals using tomography data collection software version 5.6.0.2368REL (ThermoFisher Scientific, Taipei, Taiwan), with a total dose of ~61 e^−^/Å^2^ and a defocus of −3 μm. The images had a magnification of 57,000× and were recorded with a Falcon III detector (ThermoFisher Scientific, Taipei, Taiwan).

### 3.5. Image Processing

The image processing and 3D map reconstruction of the single particle Cryo-EM data were performed with cisTEM and RELION software. The particles were picked from the middle, helical part of the WSSV nucleocapsids and extracted with a box size of 768 × 768 (55,723 particles by RELION). A total of 48,130 particles corresponding to the best 2D class averages for a narrow nucleocapsid type were selected for ab initio 3D generation using cisTEM. The generated model revealed C14 symmetry, and, then, was low-pass filtered to 60 Å for use as an initial model for 3D auto-refinement in RELION. Next, the RELION 3D classification separated the particles into two classes: a narrow type with a smaller diameter (15,279 particles) and a wider type with a larger diameter (15,243 particles). Then, 3D auto-refinement was performed, and selected classes were processed for 3D refinement by RELION. The final dataset was subjected to 3D refinement with C14 symmetry. For the cryo-electron tomography, the datasets were aligned and reconstructed using the IMOD software package version 4.10.52. All the 3D maps were displayed by UCSF Chimera. The imaging parameters are shown (Table 1). The Cryo-EM maps for the wide and narrow types of the M-NC were deposited in the Electron Microscopy Data Bank (EMDB; https://www.ebi.ac.uk/pdbe/emdb/). The EMD ID for the wide type is EMD-35598, and the narrow type is EMD-35600.

### 3.6. Ultrastructural Observations by Transmission Electron Microscopy

After the white shrimp were injected with WSSV, the stomachs were collected to investigate WSSV morphogenesis. The tissues were cut into blocks, immediately prefixed in 2.5% glutaraldehyde/0.1 M Cacodylate buffer (pH 7.4) for 4 h, 5% sucrose/0.1 M Cacodylate buffer (pH 7.4) 3 times (15 min each), and then post-fixed in 1% osmium tetroxide/0.1 M of Cacodylate buffer (pH 7.4) for 1.5 h. After ethanol dehydration, the tissue blocks were embedded in EMBed 812 resin. Ultrathin sections were prepared, stained with uranyl acetate and lead citrate, and observed with a transmission electron microscope (H7500; Hitachi, Tokyo, Japan). 

### 3.7. Localization of WSSV Major Nucleocapsid Protein VP664 on the Nucleocapsid Rings by Immunoelectron Microscopy (IEM)

Immunoelectron microscopy was performed, as described [15], with slight modifications. The samples of purified WSSV nucleocapsids were adsorbed in Formvar-supported and carbon-coated nickel grids (150 mesh) and incubated for 5~10 min at room temperature. Initially, the grids were blocked with blocking buffer (5% bovine serum albumin, 5% normal serum, 0.1% cold water skin gelatin, 10 mM of phosphate buffer, and 150 mM of NaCl, pH 7.4) for 15 min. After blocking, the grids were incubated for 1 h at room temperature with a rabbit anti-VP664 antibody diluted 1:50 in incubation buffer (0.1% Aurion Basic-c, 15 mM of NaN_3_, 10 mM of phosphate buffer, 150 mM of NaCl, pH 7.4). Next, after several washes with incubation buffer, the grids were incubated with a goat anti-rabbit secondary antibody conjugated with 12 nm diameter gold particles for 1 h at room temperature. The grids were washed several times with incubation buffer, twice more with distilled water, and stained with 2% phosphotungstic acid (pH 7.0) for 30 s. The specimens were examined with a transmission electron microscope (H7500; Hitachi, Tokyo, Japan).

## Figures and Tables

**Figure 1 ijms-24-07525-f001:**
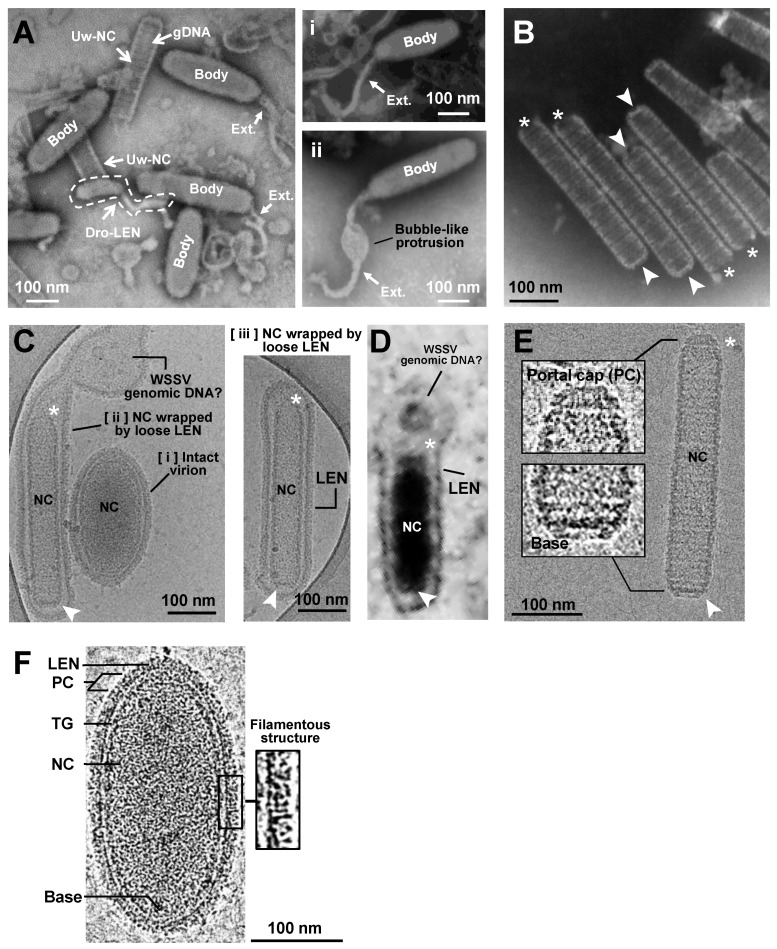
Images of WSSV enveloped virion and nucleocapsid captured with PTA-negative staining TEM and Cryo-EM. (**A**) With TEM, WSSV virions with a long oval-shaped body (Body) and with/without a flexible tail-like extension (Ext.) were observed. The dotted-line area indicates a dropped lipid-containing envelope (Dro-LEN). The nucleocapsid unwrapped by LEN (Uw-NC) indicates the prominence where the viral DNA genome (gDNAs) is aggregated. There were two types of WSSV virion with a tail-like extension (Ai,Aii). (**B**) TEM images of purified nucleocapsid without viral DNA genome (Empty NC) show a striped tube-like structure with two ends: an open portal cap end (* asterisk) and a closed basal end (arrow). (**C**) With Cryo-EM, three types of enveloped WSSV virions with the nucleocapsid (NC) were observed: [i] a shortened, oval-shaped virion with an oval-shaped NC wrapped by a firm envelope; [ii] a long virion with a striped tube NC, wrapped by a loose extra-extended envelope and an unknown aggregation (may be a WSSV genomic DNA with viral DNA-binding proteins); and [iii] a long virion with a striped tube NC wrapped by a loose envelope. The nucleocapsids had two distinct ends: the portal caps and the closed-base ends, denoted by asterisks and arrows, respectively. (**D**) TEM image of WSSV virion similar to Figure 1Cii in the nucleus of WSSV-infected cells. (**E**) With Cryo-EM, the purified striped tube-like nucleocapsid clearly had a portal cap structure (*) and closed basal end (arrow). (**F**) Cryo-EM image of the intact WSSV virion particle. The intact virion has three layers: a lipid-containing envelope (LEN), tegument (TG), and nucleocapsid (NC). The portal cap structure (PC) and closed basal end (Base) of the nucleocapsid were also observed. Filamentous structures on the surface of the lipid-containing envelope (LEN).

**Figure 2 ijms-24-07525-f002:**
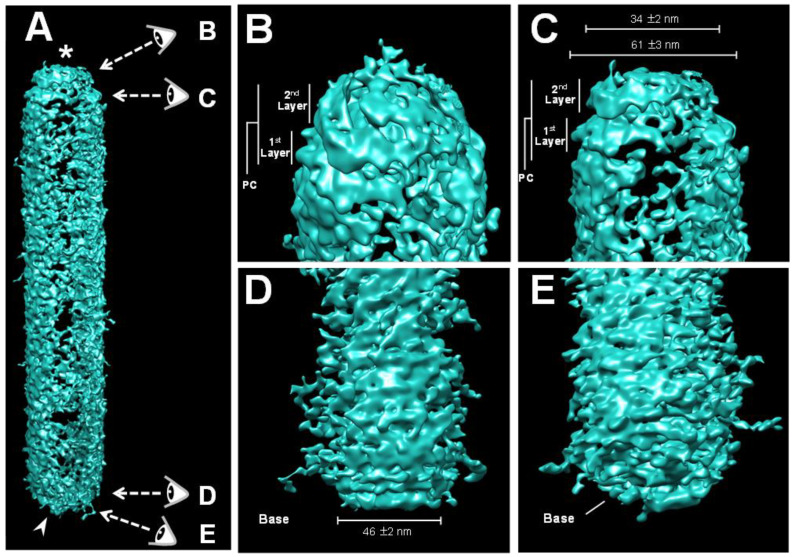
Three-dimensional volume reconstruction of WSSV nucleocapsid by cryo-electron tomography (Cryo-ET). (**A**) The reconstructed Cryo-ET of the WSSV nucleocapsid with a portal cap structure (*) and a closed basal end (arrow); the four eyes represent distinct visual angles shown in Figure (**B**–**E**); (**B**,**C**) the portal cap structure with two layers; (**D**,**E**) the closed basal end had a concave plate shape.

**Figure 3 ijms-24-07525-f003:**
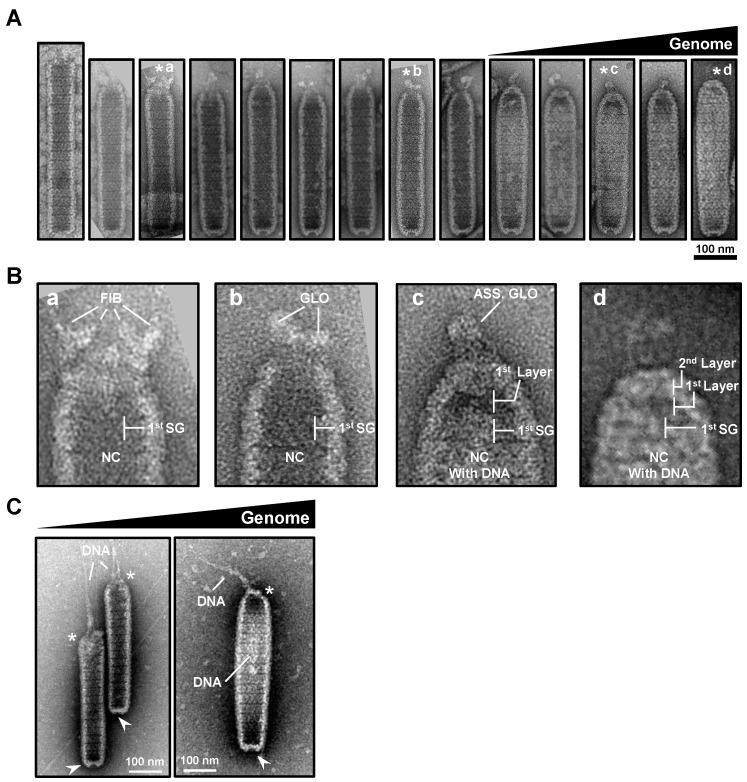
A proposed model of the WSSV nucleocapsid revealing its DNA packaging machinery based on TEM images. (**A**) Purified nucleocapsids negative-stained with UA or UF in the TEM micrographs revealed a possible process of viral DNA packaging from the opening portal cap region to form a mature intact nucleocapsid with WSSV genomic DNAs (the mist in the nucleocapsid). Potential contraction and capped events to close the opening portal region were also proposed for the nucleocapsid to assemble properly. The images were organized arbitrarily based on the cap formation, shape of the nucleocapsid, and amount of DNA packaged in the nucleocapsid. The cap ends of the nucleocapsids (labeled with asterisks) were selected in (**B**); (**B**) capped events of the opening portal region of the nucleocapsids (NC). The sequential portal cap formation is shown in **B** (**a**–**d**). First SG: the first segment of the nucleocapsid (NC); FIB: fiber-like structures; GLO: globin-like structure; ASS. GLO: assembled globin-like structure. The 1st and 2nd layers of the portal cap were designated based on Figure 2. (**C**) The potential event of WSSV DNA translocation through the portal cap end of the nucleocapsid.

**Figure 4 ijms-24-07525-f004:**
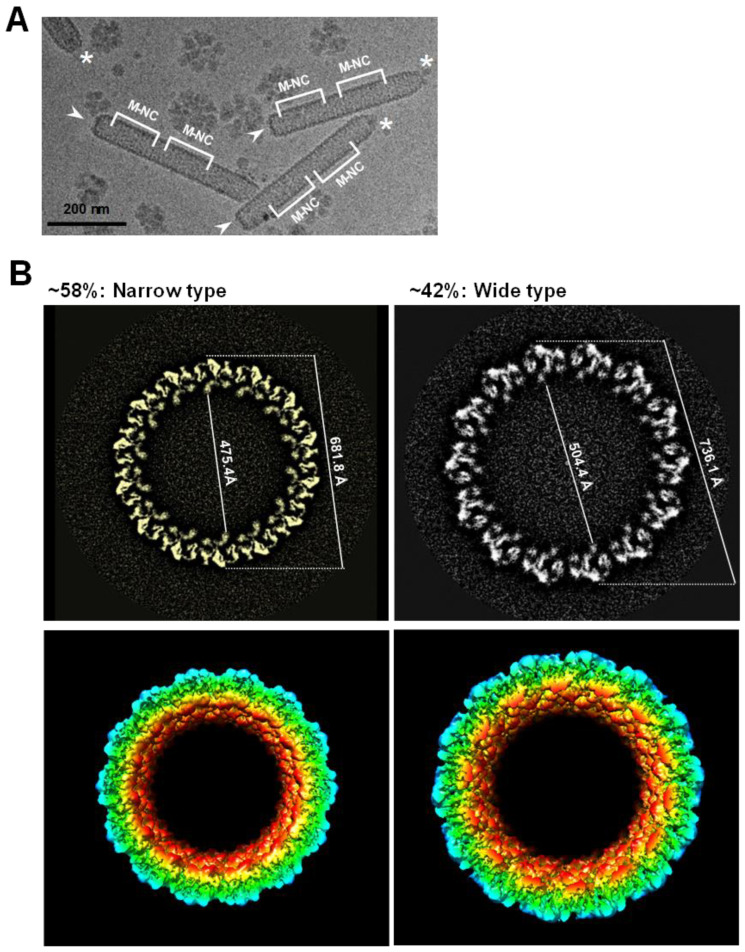
A near C14-symmetry of WSSV nucleocapsid. (**A**) Selected regions suitable for high-resolution Cryo-EM structure of WSSV nucleocapsids. After purifying WSSV nucleocapsids, the middle regions of WSSV nucleocapsid (M-NC) with relatively uniform structures were subjected to the construction of tomographic images; (**B**) diagnostic plot detail to interpret top views of the middle part of WSSV nucleocapsids (M-NC) in two packing types. Based on all the WSSV nucleocapsids evaluated, ~58% were the narrow type, whereas ~42% were the wide type. The regular feature, i.e., ‘the C14-symmetry’ of the WSSV nucleocapsid, was observed in both types. The corresponding external diameters and internal diameters of each NC type were measured.

**Figure 5 ijms-24-07525-f005:**
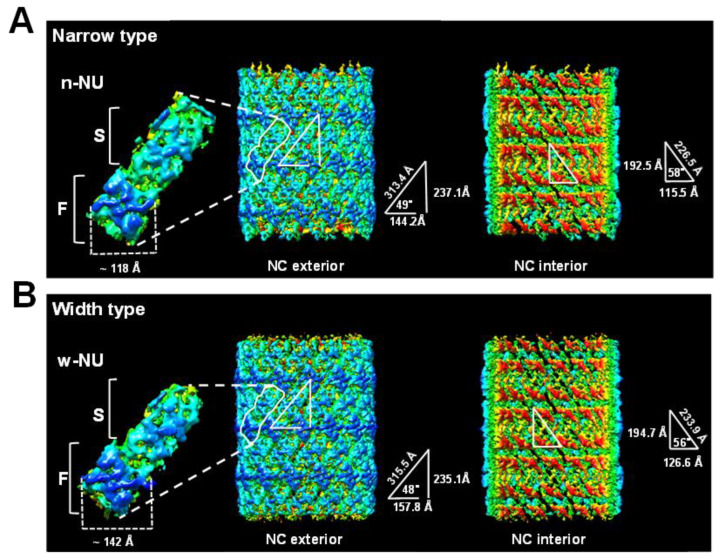
Exterior and interior surfaces of WSSV nucleocapsid (**A**) narrow type and (**B**) wide type. The left part of the images is the outer nucleocapsid surface, whereas the right part is the inner side. All the structures were radially colored from the origin (red) to the outer surface (deep blue). Nucleocapsid units (NU) with flower (F) and stem (S) domains from each type are also shown. These images were generated using CHIMERA software version 1.13.1.

**Figure 6 ijms-24-07525-f006:**
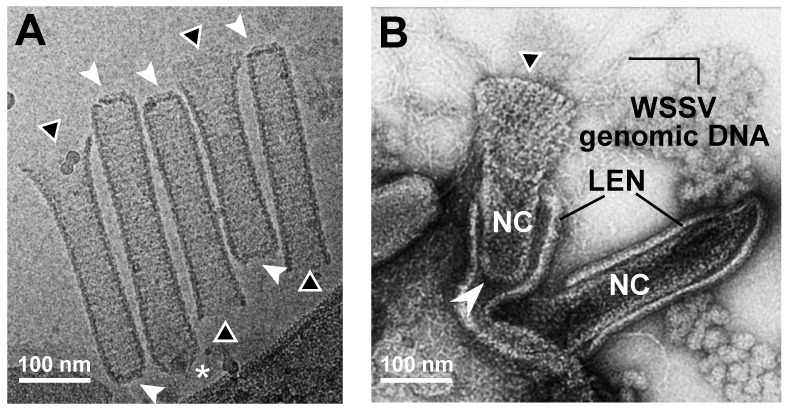
Images of dissociation of WSSV nucleocapsids. (**A**) Dissociation of WSSV nucleocapsids was initiated at the portal cap end. This image was captured with Cryo-EM; (**B**) dissociation of WSSV nucleocapsids as 14-helical strands. This image was captured with UA-negative staining TEM. *: portal cap structure; ▲: dissociated cap end; arrow: closed base end.

**Figure 7 ijms-24-07525-f007:**
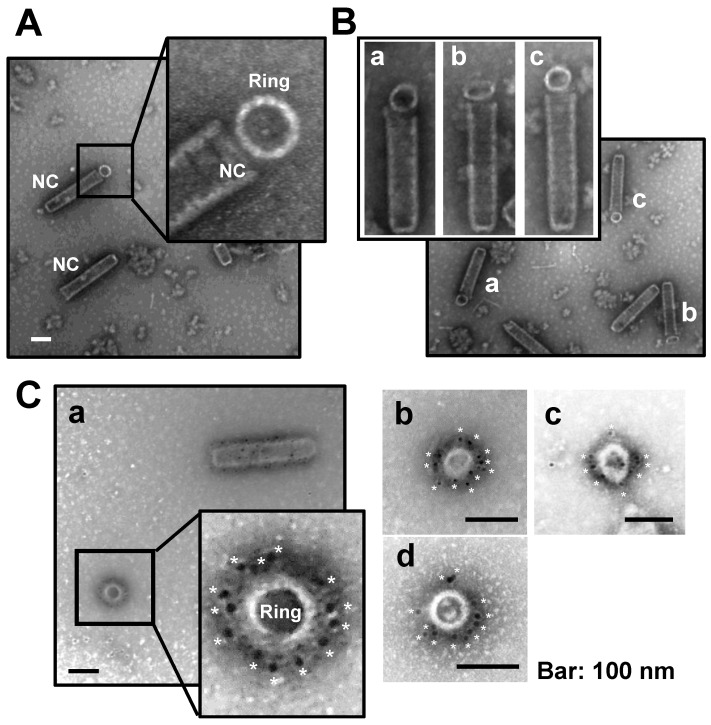
Immunoelectron microscopy analysis of purified nucleocapsid probed with a VP664 antibody. (**A**) Purified WSSV nucleocapsids had a ring-like structure. (**B**) Ring-like structures were observed in the portal cap end. (**C**) Most gold particles (asterisks) were specifically localized to the exterior side of these ring-like structures.

**Figure 8 ijms-24-07525-f008:**
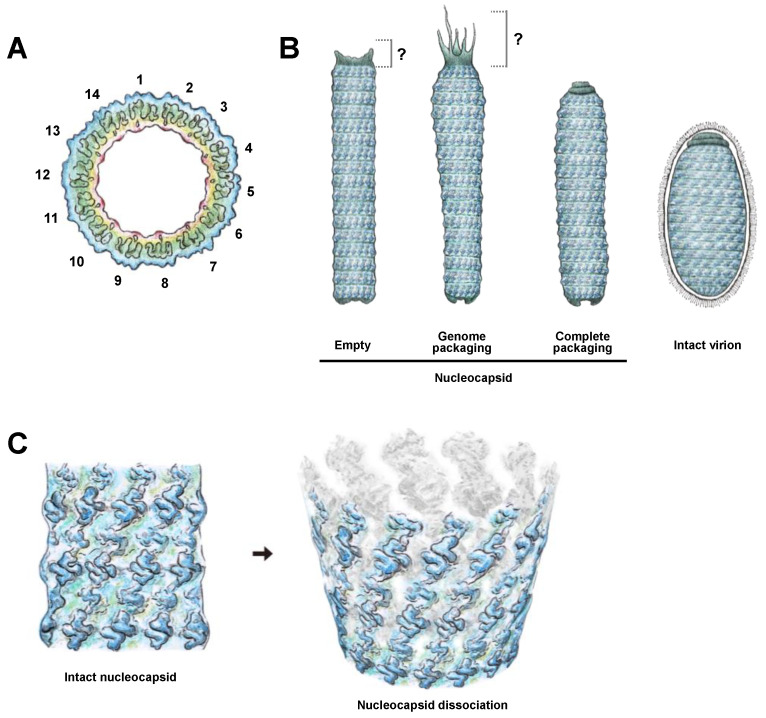
Schematic illustrations of (**A**) overall C14 symmetry of WSSV nucleocapsid; (**B**) process from empty nucleocapsids, through nucleocapsids with WSSV genomic DNAs, to intact virion with lipid-containing envelope; and (**C**) a proposed 14-helical-strand dissociation model of WSSV nucleocapsids. Note: these drawings are hand-drawn and not computer-generated.

**Figure 9 ijms-24-07525-f009:**
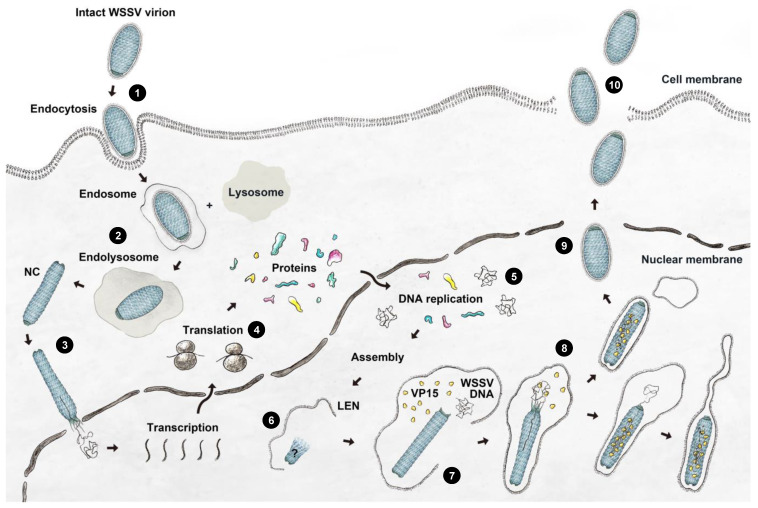
A proposed model of WSSV entry, maturation, and morphogenesis, with the following steps: (1) WSSV entry into the cell; (2) stepwise disassembly process from intact virion to nucleocapsid in endosome-lysosome fusion; (3) naked nucleocapsids presumably undergo the release of WSSV genomes from the portal cap end into the nucleus; (4) transcription and translation of WSSV genes begin when the virogenic stroma is formed; (5) replication of WSSV genomes; (6) structural proteins of envelope, tegument, and nucleocapsids start to assemble; (7) WSSV genomes with some viral DNA-binding proteins are packaged into a mature capsid through the portal cap; (8) envelope precursors begin to shrink and remove excess parts to fit the nucleocapsid; (9) intact WSSV virions are completely assembled in the nucleus; (10) intact WSSV virions are released after cell lysis.

**Table 1 ijms-24-07525-t001:** Data collection and image analysis statistics.

Type of WSSV Nucleocapsid	Narrow Type	Wide Type
Electron microscope	Talos Arctica	Talos Arctica
Detector	Falcon Ⅲ	Falcon Ⅲ
Magnification	73,000×	73,000×
Voltage(kV)	200	200
Electron exposure (e^−^/Å^2^)	48	48
Exposure (s)	2.5	2.5
Frames (no.)	50	50
Defocus range (um)	−0.5~−2.5	−0.5~−2.5
Pixel size (Å)	2.76	2.76
Symmetry imposed	C14	C14
Initial particles	55,723	55,723
Final particles	15,279	15,243
Map resolution (Å)	5.52	5.52
FSC threshold	0.143	0.143
Software	Relion and cisTEM	Relion and cisTEM

## Data Availability

The data presented in this study are available on request from the corresponding author.

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
