# Peer review of "Multiple Nucleocapsid Structural Forms of Shrimp White Spot Syndrome Virus Suggests a Novel Viral Morphogenetic Pathway"

_ijms, 2023, doi:10.3390/ijms24087525_

Round 1

Reviewer 1 Report

The manuscript "Multiple nucleocapsid structural forms of shrimp white spot syndrome virus suggest viral morphogenesis pathway" is well structured and extremely well written. The authors have undertaken a large body of work and uncovered novel insights into the structure and biology of this virus.

I only have a few typographical corrections as detailed below:

Line 39: “VP664 proteins is the main component…” correct to “VP664 proteins are the main component…”

Line 51: WSSV needs to be defined before the first use in the main body of the text, insert “(WSSV)” in the previous sentence after “white spot syndrome virus”.

Line 55: “encoded by these ORFs implicate as having….” correct to “encoded by these ORFs are implicated as having….”

Single digit numbers in the text must be written as words; please correct Line 68: "6" to "six" and Line 76: "9" to "nine".

Page 2, paragraph 4, lines 82-84: the phrase "WSSV virions" is used twice in this sentence. Please remove the second instance as it is unnecessary.

Author Response

Q1. Line 39: “VP664 proteins is the main component…” correct to “VP664 proteins are the main component…”

Ans: Thank you for pointing this out, this sentence has now been corrected (Line: 44).

Q2. Line 51: WSSV needs to be defined before the first use in the main body of the text, insert “(WSSV)” in the previous sentence after “white spot syndrome virus”.

Ans: Thank you, “(WSSV)” has now been included in Line 54.

Q3. Line 55: “encoded by these ORFs implicate as having….” correct to “encoded by these ORFs are implicated as having….”

Ans: The correction has now been included in Line 61.

Q4. Single digit numbers in the text must be written as words; please correct Line 68: "6" to "six" and Line 76: "9" to "nine".

Ans: The digit numbers in the Lines 74 and 82 has been changed to words.

Q5. Page 2, paragraph 4, lines 82-84: the phrase "WSSV virions" is used twice in this sentence. Please remove the second instance as it is unnecessary.

Ans: As suggested, the sentence has now been corrected (Line: 90).

Reviewer 2 Report

In the paper “Multiple nucleocapsid structural forms of shrimp white spot syndrome virus suggest viral morphogenesis pathway”, the authors use TEM and cryo-EM to explore the structure of white spot syndrome virus (WSSV) viral particle including the nucleocapsid structure.  Using the data, the authors propose a revised model for viral assembly and maturation.

Overall, I found the proposal to be interesting and the experiments appear well done.  However, I had issues with the interpretation of the data.  When reading the results section, I found it difficult to figure out what is experimentally verified information and what is conjecture or best guesses based on available data.  The line blurs in sections 2.4-2.6 where it seems like there is a lot of speculation without experimental verification.  I would rather see the data presented more clinically and only inferring and discussing about the specific data.  At the end of the paper, pull everything together in a hypothetical model.   That way it is clear the boundaries of what is experimentally proven and where the conjecture is.  Perhaps cut out the speculations in sections 2 and leave that to section 3.  I think the results section should stick to the points highlighted on lines 348-357 in the Conclusion.  Then expanding on lines 364-380 of the Conclusion to talk about the model and to get more speculative.  This way it is very clear to the reader what is proven and what is inferences.

Some other comments include:

·      Title of the manuscript is awkwardly phrased and unclear.  The current title doesn’t make sense “…white spot syndrome virus suggest viral morphogenesis pathway”.  Title might be revised to “…suggests a novel viral morphogenetic pathway” or something like this, but it needs revision.

·      Numerous grammar issues.  Should have an English editor review the document.  There are many examples but on line 53 “…and a least one of its capsid dimensions over 200nm”.  Should be “…and at least…”.  Makes the paper very difficult to read.

·      Figure 1B, how do the authors know that one end is open?  I can see the TEM picture with the 2 different ends, but how do we know one end is open?  Is there evidence for that or is there a way to show that it is open?

·      Seem to be using different terminology, which maybe I am misunderstanding.  But in Figure 1 the authors talk about “open cap” and “portal cap” which I believe are the same thing.  More consistent nomenclature would help for clarity. 

·      Figure 1F, if this is the terminology in the field, it is fine.  However, I found “felt-like structure” to be odd phrasing to describe a structure in an EM.

·      Line 150, it would have been helpful if the authors explained the cause of the lack of tail-like extensions were observed under cryo-EM.  Is this an artefact of sample preparations?  Why does it appear it some instances (Figure 1A) but not in others (Figure 1F).  I know more info was provided on lines 157-160, but I still find this issue confusing as described.

·      One of the issues I have is how do we know what we are looking at?  For example, lines 214-217.  There are a lot of assumptions without proof here.  How do we know that DNA packaging is occurring?  Figure 3C points out DNA, but how do we know that is DNA and not some protein fiber?  Perhaps this is well known in the field, but it feels like assumptions about structures is being made in the paper without backing it up with proof that is what is happening.  

·      Lines 214-215 talk about portal cap-like structure was formed gradually in the apical direction.  I don’t really see in the images what they are referring to.  Also how do we know that we are looking at successive or sequential stages and not a variety of artifacts lined up?  Paper seems to be drawing mechanistic conclusions from a series of static images and how do we know they are lined up in the correct order.

·      Lines 218-224, like the previous comments.  Inferences to what is happening is being made, but I am not convinced that we know the sequence of events.  Are the authors lining things up to support a particular narrative vs this is the actual staging.  How can the authors show which stages are first and which are last.  Perhaps using blockers of DNA replication, to prevent packaging to identify the starting stages.  I feel like more mechanisms are being drawn here then the data supports.  To restate how do we know in Figure 3A, that the 4th image in the panel comes before the 5th, 6th, 7th, and 8thimage.  Couldn’t they be arranged in a different order?  If not, why not.  If the authors do have a sense for the proper order of the images, this should be outlined in the results to give the reader confidence that the model being drawn is consistent with the actual stages in vivo.

Author Response

Q1. Title of the manuscript is awkwardly phrased and unclear.  The current title doesn’t make sense “…white spot syndrome virus suggest viral morphogenesis pathway”.  Title might be revised to “…suggests a novel viral morphogenetic pathway” or something like this, but it needs revision.

Ans: Thank you for the suggestion. The title has now been changed to “Multiple nucleocapsid structural forms of shrimp white spot syndrome virus suggests a novel viral morphogenetic pathway”

Q2. Numerous grammar issues.  Should have an English editor review the document.  There are many examples but on line 53 “…and a least one of its capsid dimensions over 200nm”.  Should be “…and at least…”.  Makes the paper very difficult to read.

Ans: Thank you for pointing this out. The grammatical corrections have now been included in Line 56 and elsewhere in the MS. The MS was further edited by a native English speaker Prof. John Kastelic, University of Calgary, Canada.

Q3. Figure 1B, how do the authors know that one end is open?  I can see the TEM picture with the 2 different ends, but how do we know one end is open?  Is there evidence for that or is there a way to show that it is open?

Ans: The open end of the nucleocapsid is designated based on the dissociation of the cap at one end while the basal end is completely closed. In addition, we can see the genomic DNA and the loose envelope associated with the open end as seen in Fig. 1C and 1D. Moreover, a recent publication (Sun et al., 2023; [Fig. 4]) also pointed out that the open end (head) is involved in releasing genomic DNA.

Q4. Seem to be using different terminology, which maybe I am misunderstanding.  But in Figure 1 the authors talk about “open cap” and “portal cap” which I believe are the same thing.  More consistent nomenclature would help for clarity. 

Ans: To keep the terminology consistent, the “open cap” has now been changed to “opened portal cap” in the MS

Q5. Figure 1F, if this is the terminology in the field, it is fine.  However, I found “felt-like structure” to be odd phrasing to describe a structure in an EM.

Ans: The terminology “felt-like structure” has now been changed to “filamentous structure” in Fig. 1F and also in Lines 166 and 197.

Q6. Line 150, it would have been helpful if the authors explained the cause of the lack of tail-like extensions were observed under cryo-EM.  Is this an artefact of sample preparations?  Why does it appear it some instances (Figure 1A) but not in others (Figure 1F).  I know more info was provided on lines 157-160, but I still find this issue confusing as described.

Ans: Thank you this question. As you can see in Fig 1C, the NC is wrapped by loose LEN (an envelope surplus), during TEM sample preparation, the dehydration step leads to the constriction of the loose LEN and makes it appear as tail-like extensions. Since the cryo-EM procedure retains water molecule in the sample, a mature WSSV virion has no tail-like extension and the envelope is tightly wrapped around the nucleocapsid. Therefore, the tail-like structures appear only in TEM images and not in cryo-EM images. The same has now been included in Lines 159-162.

Q7. One of the issues I have is how do we know what we are looking at?  For example, lines 214-217.  There are a lot of assumptions without proof here.  How do we know that DNA packaging is occurring?  Figure 3C points out DNA, but how do we know that is DNA and not some protein fiber?  Perhaps this is well known in the field, but it feels like assumptions about structures is being made in the paper without backing it up with proof that is what is happening.  

Ans: Thank you for raising this question. We agree to your point, at present we do not have an experimental proof to confirm the DNA molecule. Due to the lack of shrimp cell line, we tried expressing the WSSV nucleocapsid in the insect Sf9 cells but failed. However, based on our previous publication (Tsai et al., 2006) and a recent publication (Sun et al., 2023) we strongly believe that it is the DNA that is released from the nucleocapsid and not a protein fiber.

[Tsai et al., Identification of the nucleocapsid, tegument, and envelope proteins of the shrimp white spot syndrome virus virion. J. Virol. 2006, 80, 3021-3029.

Sun et al., Ring-stacked capsids of white spot syndrome virus and structural transitions with genome ejection. Sci. Adv. 2023, 9, eadd2796]

Q8. Lines 214-215 talk about portal cap-like structure was formed gradually in the apical direction.  I don’t really see in the images what they are referring to.  Also how do we know that we are looking at successive or sequential stages and not a variety of artifacts lined up?  Paper seems to be drawing mechanistic conclusions from a series of static images and how do we know they are lined up in the correct order.

Ans: We agree that we have arranged a series of static images in Fig 3A representing the DNA packaging. However, these images were arranged based on (i) the stages of cap formation; (ii) the change in the shape of the nucleocapsid from thin-rod shaped to shorter fat-rod shaped; (iii) the amount of DNA packaged in the nucleocapsid (successive increase in DNA amount). In addition, our data arrangement is also based on the DNA packaging reported in Baculovirus (Wang et al., 2016; Zhao et al., 2019). The same has now been included in Lines 231-234 and 268-270.

[Wang et al., Budded baculovirus particle structure revisited. J. Invertebr. Pathol. 2016, 134, 15-22

Zhao et al., Nucleocapsid Assembly of Baculoviruses. Viruses 2019, 11, 595]

Q9. Lines 218-224, like the previous comments.  Inferences to what is happening is being made, but I am not convinced that we know the sequence of events.  Are the authors lining things up to support a particular narrative vs this is the actual staging.  How can the authors show which stages are first and which are last.  Perhaps using blockers of DNA replication, to prevent packaging to identify the starting stages.  I feel like more mechanisms are being drawn here then the data supports.  To restate how do we know in Figure 3A, that the 4th image in the panel comes before the 5th, 6th, 7th, and 8thimage.  Couldn’t they be arranged in a different order?  If not, why not.  If the authors do have a sense for the proper order of the images, this should be outlined in the results to give the reader confidence that the model being drawn is consistent with the actual stages in vivo.

Ans: Please refer to the response to Reviewer 2 Q8. The sense of proper order and the strategy for arranging the Fig 3A has now been included in the results section (Lines: 231-234) and the figure legend for Fig 3 (Lines: 268-270)

Reviewer 3 Report

The authors utilized electron microscopy to characterize the WSSV virion and the nucleocapsid, providing interesting new insights into the virion morphology.  

 Some concerns and suggestions: 

1.       For section 2.1, please discuss the potential reasons for different observations under ns TEM and cryoEM. In line 108, a mature virion with a tail-like extension is referred to as one of the two virion forms. And in line 157, it is concluded that the virion with envelope surplus is immature. Please explain why it is concluded that the mature form doesn’t have a tail-like structure. Please also consider including a discussion on the proposed nature/function/(non-) importance of the tail-like structure. 

2.       Figure 3B: scale bar

3.       Line 320: Quote “some ring-like structures closed to the portal cap end were observed or individually”. Please edit.

4.       Figure 7C: please consider adding labels to the figure to help illustrate the model.

5.       In the discussion section starting from Line 364: in the proposed morphogenesis model, please highlight the new additions to the previous model based on the data in this paper.

6.       In the discussion section starting from Line 391: Please be more specific on what the information or knowledge is referred to. 

Author Response

Q1. For section 2.1, please discuss the potential reasons for different observations under ns TEM and cryoEM. In line 108, a mature virion with a tail-like extension is referred to as one of the two virion forms. And in line 157, it is concluded that the virion with envelope surplus is immature. Please explain why it is concluded that the mature form doesn’t have a tail-like structure. Please also consider including a discussion on the proposed nature/function/(non-) importance of the tail-like structure. 

Ans: Thank you the question. Please refer to the response to Reviewer 2 Q6. This explanation has now been included in Lines: 159-162.

Q2. Figure 3B: scale bar

Ans: Figure 3B is an enlarged image of Fig. 3A which is designated with ‘*’ and alphabets ‘a’ ‘b’ ‘c’ and ‘d’. Hence, we did not include the scale bar for Fig. 3B.

Q3. Line 320: Quote “some ring-like structures closed to the portal cap end were observed or individually”. Please edit.

Ans: The sentence has now been corrected to “we noticed some ring-like structures associated with portal cap end of the nucleocapsid (Fig. 7B) as well as individual ring-like structures (Fig. 7C).” in Lines: 367-369.

Q4. Figure 7C: please consider adding labels to the figure to help illustrate the model.

Ans: The labels “Intact nucleocapsid” and “Nucleocapsid dissociation” has now been added to Fig. 8C. Please note that the figure numbers have now been changed, hence Fig. 7C now refers to 8C.

Q5. In the discussion section starting from Line 364: in the proposed morphogenesis model, please highlight the new additions to the previous model based on the data in this paper.

Ans: As suggested the new additions to the previous model is included in Lines: 432-433, 438-440 and 441-442.

Q6. In the discussion section starting from Line 391: Please be more specific on what the information or knowledge is referred to. 

Ans: The information in the conclusion section has now been edited as per your suggestion. (Lines: 465-471)